

# Iron and copper on *Botrytis cinerea*: new inputs in the cellular characterization of their inhibitory effect

Fátima Rodríguez-Ramos[1], Vilbett Briones-Labarca[1], Verónica Plaza[2] and Luis Castillo[2]

[1] Departamento de Ingeniería en Alimentos, Universidad de La Serena, La Serena, Coquimbo, Chile
[2] Departamento de Biología, Universidad de La Serena, La Serena, Coquimbo, Chile

## ABSTRACT

Certain metals play key roles in infection by the gray mold fungus, *Botrytis cinerea*. Among them, copper and iron are necessary for redox and catalytic activity of enzymes and metalloproteins, but at high concentrations they are toxic. Understanding the mechanism requires more cell characterization studies for developing new, targeted metal-based fungicides to control fungal diseases on food crops. This study aims to characterize the inhibitory effect of copper and iron on *B. cinerea* by evaluating mycelial growth, sensitivity to cell wall perturbing agents (congo red and calcofluor white), membrane integrity, adhesion, conidial germination, and virulence. Tests of copper over the range of 2 to 8 mM and iron at 2 to 20 mM revealed that the concentration capable of reducing mycelial growth by 50% ($IC_{50}$) was 2.87 mM and 9.08 mM for copper and iron, respectively. When mixed at equimolar amounts there was a significant inhibitory effect mostly attributable to copper. The effect of $Cu_{50}$, $Fe_{50}$, and $Cu_{50}$–$Fe_{50}$ was also studied on the mycelial growth of three wild *B. cinerea* strains, which were more sensitive to metallic inhibitors. A significant inhibition of conidial germination was correlated with adhesion capacity, indicating potential usefulness in controlling disease at early stages of crop growth. Comparisons of the effects of disruptive agents on the cell wall showed that Cu, Fe, and Cu–Fe did not exert their antifungal effect on the cell wall of *B. cinerea*. However, a relevant effect was observed on plasma membrane integrity. The pathogenicity test confirmed that virulence was correlated with the individual presence of Cu and Fe. Our results represent an important contribution that could be used to formulate and test metal-based fungicides targeted at early prevention or control of *B. cinerea*.

# INTRODUCTION

*Botrytis cinerea* is a phytopathogen of great interest in the agri-food field. As the second most critical fungal disease worldwide (*Dean et al., 2012*), and one that can affect a broad range of plant species (*Kai et al., 2020*), *B. cinerea* causes significant pre- and post-harvest losses, estimated at US\$10–100 billion annually worldwide (*Hua et al., 2018*). It is capable of infecting more than 500 species of vascular plants (*Spada et al., 2021*), predominantly fruits and vegetables, affecting all organs: flowers, leaves, roots, and shoots (*Torres-Ossandón*

Corresponding author
Luis Castillo, lcastillo@userena.cl

*et al., 2019*). The main affected crops include vegetables such as tomato, zucchini, and cucumber, and fruit-bearing plants like grape, strawberry, and raspberry (*Cheung et al., 2020*).

Chemical fungicides are the most common biocides used to control *B. cinerea* (*Castro et al., 2019*; *Nakajima & Akutsu, 2014*). Hydroxyanilides, anilinopyrimidines, carboxamides, phenylpyrroles, and benzimidazole carbendazim are the fungicides widely used against this pathogen (*Kai et al., 2020*; *Olea et al., 2019*). They have diverse mechanisms of action. For example, fenhexamid inhibits the 3-ketoreductase involved in demethylation during ergosterol biosynthesis (*Debieu et al., 2013*), while boscalid is a succinate dehydrogenase inhibitor (*Cherrad et al., 2018*). Indiscriminate chemical use has resulted in the appearance of *Botrytis* strains resistant to a single fungicide or multiple fungicides. In addition, the fungicide compounds could cause harm to human health and the environment (*Rupp et al., 2017*). The numerous enzymes, metalloproteins, and metabolites produced by *B. cinerea* may play key roles in the infection and proliferation of the pathogen (*Nakajima & Akutsu, 2014*; *Choquer et al., 2007*). Transition metals provide the necessary redox and catalytic activity for various biological processes and may have direct or indirect roles in the pathogenicity of *B. cinerea* (*Gerwien et al., 2018*). These processes, require the ubiquitous presence of metal ions, such as Co, Cu, Mn, Fe, and Ni, as catalysts, structural elements in proteins, for electron transfer reactions, or as messengers (*Antsotegi-Uskola, Markina-Iñarrairaegui & Ugalde, 2020*). However, although metals are essential elements for growth and survival, both deprivation as well as metal overload is detrimental (*Blatzer & Latgé, 2017*).

Copper and iron have been studied to elucidate their role in the virulence pathways of fungi (*Antsotegi-Uskola, Markina-Iñarrairaegui & Ugalde, 2020*; *Gerwien et al., 2018*). In the specific case of copper, two oxidation forms are usually present in the environment: $Cu^{2+}$ and $Cu^+$, but only the form 1+ is recognized as a substrate by the transporters, since the extracellular $Cu^{2+}$ is reduced *via* plasma membrane reductases prior to incorporation. $Cu^+$ uses a passive transport to enter the cell, helped by an extremely low intracellular copper concentration (*Balamurugan & Schaffner, 2006*). When the intracellular copper levels exceed the toxicity threshold in fungi, detoxification mechanisms are activated in order to reestablish cellular copper balance: the first mechanism is related to the chelation due copper sequestration by metallothionein's, and the second mechanism relies on PIB-type ATPases, which contain metal (Cu) binding domains rich in cysteine and histidine (*Antsotegi-Uskola, Markina-Iñarrairaegui & Ugalde, 2020*). In the case of *B. cinerea*, copper-dependent proteins play an essential role in many aspects of virulence, including pathogenesis and the copper detoxification system, as critical factors in the infection mechanism (*Saitoh et al., 2010*; *Smith, Logeman & Thiele, 2017*).

Iron is an essential nutrient for all eukaryotes and almost all prokaryotes due to its required metabolic function (*Jhonson, 2008*). It has the most diverse roles in cellular processes, especially in central metabolic pathways such as oxygen delivery and the electron transport chain, primarily *via* incorporation of iron into the active centers of crucial enzymes, or in the utilization of iron in siderophores (*Gerwien et al., 2018*). This metal is considered a fundamental element for virulence in many microbial pathogens,

including *B. cinerea* (*Vasquez-Montaño et al., 2020*). Most fungi and bacteria exhibit specific mechanisms for the acquisition of iron from the hosts they infect for their own survival, because iron is not easily available in the environment due to host sequestration (*Jhonson, 2008*). In the case of fungi, four different mechanisms have been identified: (1) ferric iron uptake mediated low- molecular-weight Fe (III)-siderophores, which are non-ribosomally synthesized secreted iron chelators key in the uptake, intracellular transport, and storage of iron (*Haas, 2014*); (2) reductive iron assimilation (RIA) through a plasma membrane system; (3) low-affinity ferrous iron uptake and, (4) heme uptake and degradation (*Haas, 2014*; *Condon et al., 2014*; *Vasquez-Montaño et al., 2020*). Most fungal species employ more than one of these systems in parallel but rarely are all four strategies present in the same specie (*Haas, Eisendle & Turgeon, 2008*). Specifically, the siderophores have been considered relevant not only for adaptation to iron starvation conditions or iron storage, but also for asexual and sexual propagation, germination, resistant to oxidative stress, mutual interaction, protection against iron-induced toxicity in some fungal organisms, microbial competition as well as virulence in plant and animal hosts (*Jhonson, 2008*; *Haas, 2014*). High-affinity iron uptake systems function during iron-limiting conditions, such as siderophore-mediated iron uptake and reductive iron assimilation, enable fungi to acquire limited iron from animal or plant hosts, whereas low-affinity systems are important during periods of relative iron abundance (*Jhonson, 2008*), and include the iron-containing protein (*e.g.*, heme, ferredoxin) uptake pathway and the ferrous iron uptake pathway, where the siderophore-mediated iron uptake pathway is the most extensively reported (*Liu et al., 2020*). Regarding *B. cinerea*, is expected to produce at least nine siderophores, being ferrirhodin the main and the most abundant, but this pathogen can also take up five other known siderophores, with the following order of uptake: ferrichrysin, ferrirubin, ferrirhodin, hexahydroferrirhodin and, coprogen (*Vasquez-Montaño et al., 2020*; *Konetschny-Rapp et al., 1988*).

As redox-active metals, the redox properties of iron and copper (variable oxidation states, catalytic activity, complex formation, redox couples), are also responsible for the toxicity of these elements when they are in excess because of the formation of highly reactive oxygen species through the Fenton reaction. In addition, copper can displace iron in various proteins, emphasizing the need for an optimal balance of these metals (*Blatzer & Latgé, 2017*). Since copper at high enough concentration is toxic to fungi, a variety of copper compounds has been used through the years as antifungal agents in agriculture. Since the first use of copper in vineyards in 1885, through the well-known Bordeaux mixture ($CuSO_4 \cdot 5H_2O + Ca (OH)_2$) for the control of pathogenic fungi such as *Plasmopara viticola* and *B. cinerea* (*Judet-Correia et al., 2011*), there has been an evolution in the development of antimicrobial formulations based on copper for the protection of crops. Currently, there are several copper-based fungicides commercially available: "Cobre premium" (copper oxide (I), 60%) from Syngenta®; Agrocopper ($CuSO_4 \cdot 5H_2O$, 98%) from Bayer®; and Captain Jacks® based on $C_{16}H_{30}CuO_4$ (10%) from Bonide.

*Judet-Correia et al. (2011)* modeled the inhibitory effect of Cu on the growth of *B. cinerea* (strains BC1 and BC2), where the half-maximal inhibitory concentration ($IC_{50}$) was 2.21 and 2.60 mM, for the strains respectively, indicating that Cu may bind to the surface
of the spores during germination, requiring adequate time for a detoxification process and a subsequent selection of surviving spores. In general, the mechanism of action of Cu in fungi has not been entirely elucidated: low concentrations of this metal act as an essential nutrient facilitating its proliferation, while higher concentrations are fungitoxic. Its toxicity is related to three relevant effects: (1) delays in the germination of spores; (2) redox imbalance at the cellular level with damage to membranes; and (3) inhibitory effects on the activity of the enzyme laccase, preventing infection (*Buddhika, Savocchia & Steel, 2020*; *Judet-Correia et al., 2011*; *Lamichhane et al., 2018*). Iron, which can exist as ferrous ($Fe^{2+}$) or ferric ($Fe^{3+}$) ions, is an essential nutrient and necessary for the virulence of fungi that cause disease (*Robinson, Isikhuemhen & Anike, 2021*). Unlike Cu, Fe has been widely applied as a fertilizer. Since this metal is considered the third most limiting nutrient for plant growth and metabolism and, its deficiency is a usual nutritional disorder in many crops, resulting in poor yields and reduced nutritional quality (*Rout & Sahoo, 2015*). Specifically, the use of iron as foliar fertilizer has often affected positively the yields and quality of fruits and other crops, especially when levels of available soil nutrients are low (*Ma et al., 2019*). However, various studies have evaluated its effect as an antifungal at certain concentrations, because of the reported triggering at the cellular level of reactive oxygen species *via* Fenton chemistry, resulting in oxidative damage to the microbes (*Gerwien et al., 2018*; *Touati, 2000*). Studies have been done to assess the effect of Fe on *B. cinerea* and other relevant phytopathogens in grape crops (*Fleurat-Lessard et al., 2011*), and 10 mM caused dramatic changes in hyphal organization, favoring cell death, while no toxicity was observed on grapevine leaves.

Previous evidence showed that iron and copper compounds could successfully inhibit mycelial growth and conidial germination of *B. cinerea*. However, more studies are needed to expand our understanding of the mechanisms of action at the cellular level for this critically important pathogen. Thus, this study aimed to deliver new contributions in this area by characterizing the antifungal effect of copper, iron, and their mixture on *B. cinerea*, evaluating membrane integrity, cell wall alteration, adhesion capacity, conidial germination, and pathogenicity.

## MATERIALS & METHODS

### Fungal strains and growth conditions

The *B. cinerea* reference strain B05.10, a haploid wild-type strain originally isolated from grapes (*Biittner et al., 1994*; *Quidde, Osbourn & Tudzynski, 1998*) is available from the International Collection of Microorganisms from Plants (New Zealand, Strain no. ICMP 17.435). The three wild strains studied (Bc.po03, Bc. vi09, and Bc. ad03) were isolated from different plants and locations in Coquimbo Region, Chile (B Marambio, L Olivares, V Plaza, L Aguilera, L Castillo, 2020, unpublished data) (See Table S1). *B. cinerea* strains were grown on potato dextrose agar (PDA; Difco™), malt extract broth (MEB), and malt extract agar (MEA, 2% malt extract and 2% agar, Merck). MEA plates were inoculated with 10 µL of conidial suspension ($2.5 \times 10^5$ conidia/mL) and cultured for 12 days at 25 °C under a photoperiod of 12 h light/12 h darkness in a Percival incubator (Percival Scientific,

Perry, IA, USA) followed by 10 days of total darkness. Sporulating cultures were washed with sterile water and filtered (*Plaza et al., 2015*).

## DNA extraction, PCR amplification and sequencing

Genomic DNA extraction was performed as in a previous publication according *Notte et al. (2021)* and it was using standard protocols (*Sambrook, Fritsch & Maniatis, 1989*). The DNA pellets were dissolved in 50 µL of TE buffer (10 mM Tris–HCl [pH 8.0], 1mM EDTA) and quantified using a Nanodrop spectrophotometer. To identify *B. cinerea* species we sequenced the ITS region and glyceraldehyde-3-phosphate dehydrogenase (G3PDH), Heat-shock Protein 60 (HSP60), DNA-dependent RNA polymerase subunit II (RPB2) and protein encoding necrosis and ethylene inducing protein 1 (NEP1) genes (*Staats, Baarlen & Van Kan, 2005*; *White et al., 1990*). Primer pairs (Table S2) were designed to amplify the *G3PDH*, *HSP60*, *RPB2* and *NEP1* genes. For amplifications of ITS region, primers IST1 and ITS4 were used (*White et al., 1990*). The PCR reaction and the thermocycling pattern to amplify *HSP60, G3PDH,* NEP1 and *RPB2* were following according to *Notte et al. (2021)* and *Staats, Baarlen & Van Kan (2005)*. The presence of transposons was detected by PCR with the primers boty-F and boty-R to amplify boty transposon, and the primers F300 and F1500 to detect the flipper transposon (Table S2) according to *Giraud et al. (1999)* and *Muñoz et al. (2010)*, respectively.

## Molecular and phylogenetic identification

For the identification of native isolates *Botrytis* sp., analysis of *HSP60*, *RPB2*, *G3PDH* and *NEP1* genes were sequenced using the same primers by Macrogen (Seoul, Korea). The multiple alignments were performed using CLUSTAL W (*Thompson, Higgins & Gibson, 1994*). The phylogenetic trees were constructed based on the neighbor-joining method (*Saitou & Nei, 1987*) and, the topology confirmed with the Maximum Likelihood and Maximum Parsimony methods by using the MEGA X software (*Kumar et al., 2018*). Distance matrices were calculated by the Kimura 2-parameter method and bootstrap analysis was performed based on 1,000 re-samplings (*Kumar et al., 2018*). Nucleotide sequences of *HSP60*, *RPB2*, *G3PDH* and *NEP1*, commonly used for differentiation of *Botrytis* species, were obtained from the GenBank website and compared with the partial sequences of the HSP60 (705 bp), RPB2 (924 bp), G3PDH (690 bp) and NEP1 (336 bp) genes from native isolates (Bc.ad03, Bc.po03 and Bc.vi09) (Table S4).

## Effect of Cu, Fe, and Cu–Fe on mycelial growth of B05.10 and wild strains of *B. cinerea*

The effect of copper and iron on the mycelial growth of B05.10 was measured using the radial growth test on PDA (pH 5.40). The methodology was based on a modified version of the protocol developed by *Judet-Correia et al. (2011)* and *Fleurat-Lessard et al. (2011)*. Stock solutions of Cu (100 mM [pH 2.01]), and Fe (200 mM [pH 2.34]) ($CuSO_4 \cdot 5H_2O$ and $FeSO_4 \cdot 7H_2O$, Merck) in sterile water were prepared, filtered through an MF-Millipore™ membrane filter (0.22 µm pore size), and diluted to obtain a range of final concentrations in medium (Cu: 2, 4, 6, 8 mM; Fe:2, 4, 6, 8, 10, 12, 14, 20 mM; and equimolar mixture of Cu–Fe: 2, 4, 6, 8, and 12 mM). The medium was poured into 90 mm diameter Petri

dishes, which were inoculated with 10 µL of conidial suspension ($1 \times 10^6$ conidia/mL) and incubated at 25 °C in the dark for seven days. Mycelial growth diameters were measured daily using a caliper, and the inhibition values were calculated. $IC_{50}$ values for B05.10 mycelial growth were estimated at day 7 of incubation and analyzed by linear regression using the software GraphPad Prism (version 9.2.1). All experiments were performed with $n = 6$ samples and the results were expressed as means ± SD. The same procedure previously described, was used to measure the effects of metal inhibitors on the mycelial growth of three wild strains of *B. cinerea* (Table S1) for 6 days, but considering the $IC_{50}$ values estimated previously, and the inhibition rate was compared with the B05.10 strain.

The inhibitory effect on *B. cinerea* (B05.10 and wild strains) was also studied in the presence of mixtures based on metallic inhibitors Cu ($IC_{50}$ 2.87 mM) and Fe ($IC_{50}$ 9.08 mM), with commercial fungicides (12.8 ppm boscalid, 4.8 ppm iprodione, and 15.35 ppm fenhexamid) in terms of the survival rate (%). For this purpose, a preliminary analysis of chemical compatibility in terms of solubility was initially performed for the antifungal-metal mixtures to define the antifungals to be studied. The assay was conducted in 96-well plates using MEB medium, considering the study of the mixtures and controls: antifungals, metals, and strains under study. In each well, a total of 100 µL of each variant to be evaluated was added, followed by 100 µL of conidial suspensions ($2.5 \times 10^5$ conidia/mL) and incubated at 25 °C for 72 h, for a total volume of 200 µl, ensuring the desired concentrations to be evaluated. Finally, the absorbance at 600 nm was measured with an Epoch microplate spectrophotometer (BioTek Instruments, Winooski, VT, USA). The optical density (OD) values were proportional to the quantity of biofilm formed, comprising mycelium and extracellular polymeric material. Results for the OD were compared with the control to estimate the survival rate of the variants studied. The synergism pattern among the components of mixtures was determined according to the mixture index (MI) expression (*Giovagnoli-Vicuña et al., 2019*):

$$MI = \frac{AC_1C_2}{AC_1 + AC_2}$$

where $AC_1C_2$ was the value of inhibition percentage (%) for mixtures and ($AC1 + AC2$) was the value obtained by the sum of % inhibition for each individual inhibitor (antifungal or metal). The following cutoff values were chosen, for the interpretation of obtained results: synergism MI is >1, MI = 1 addition, and MI<1 would be antagonism.

### Effect of Cu, Fe, and Cu–Fe on adhesion capacity of B05.10.

The adhesion capacity of *B. cinerea* was measured in the presence of $Cu_{50}$, $Fe_{50}$, and the mixture ($Cu_{50}$–$Fe_{50}$) to determine their effect on this parameter. The methodology used was based on a modified version of the protocol developed by *Plaza et al. (2015)*. First, aliquots of 100 µL of $2.5 \times 10^5$ conidia/mL were added to sterile polystyrene 96-well microtiter plates (JetBiofil, China). Then, 200 µL of Malt Extract Broth (2% Malt) (MEB, pH 4.70) containing the concentrations of $Cu_{50}$ (pH 3.45), $Fe_{50}$ (pH 3.88), and $Cu_{50}$–$Fe_{50}$ (pH 3.33) were added to reach the $IC_{50}$ values in each well. Plates were incubated in darkness at 25 °C for five days, then the medium containing unbound conidia was aspirated, and remaining non-adherent conidia were removed by washing three times with 250 µL sterile

water. Next, 100 μL of 0.1% methyl violet in glacial acetic acid was added to the adherent cells and allowed to stain for 5 min. The wells were washed three times with 250 μL of phosphate-buffered saline (PBS) and absorbance at 590 nm was measured with an Epoch microplate spectrophotometer (BioTek Instruments, USA). The optical density (OD) values were proportional to the quantity of biofilm formed, comprising mycelium and extracellular polymeric material.

### Effect of Cu, Fe, and Cu–Fe on conidial germination of B05.10

The effect on conidial germination of *B. cinerea* in the presence of $Cu_{50}$, $Fe_{50}$, or $Cu_{50}$–$Fe_{50}$ was determined according to the methodology of *Torres-Ossandón et al. (2019)*. Briefly, 500 μL of an aqueous suspension of $2.5 \times 10^5$ conidia/mL and aliquots of inhibitors were combined and the volume made up to 1.0 mL with MEB medium, and incubated at 22 °C using constant agitation for 0, 4, and 6 h. Afterwards, the conidia were collected by centrifugation at $13.680 \times g$ for 5 min and washed twice with 500 μL of sterile water. The pellet was resuspended in 100 μL of sterile water and 10 μL aliquots of each sample were placed on slides, cover-slipped, and examined under a light microscope (Eclipse E-200). The percentage (%) of germinated conidia was assessed on microphotographs of the images. As a rule, the conidia were considered germinated when the length of the germ tube was twice the length of the conidia (*Torres-Ossandón et al., 2019*). Three replicates were evaluated for each treatment, and a minimum of 100 conidia was counted for each replicate.

### Membrane integrity assay

The effect on *B. cinerea* membrane integrity in the presence of $Cu_{50}$, $Fe_{50}$, and $Cu_{50}$–$Fe_{50}$ was determined according to the methodology of (*Torres-Ossandón et al., 2019*). Aqueous suspensions of $5 \times 10^5$ conidia/mL untreated and treated with inhibitors in MEB medium (final volume of 1 mL) were incubated in a shaker at 22 °C for 4 and 6 h. Afterwards, the conidia were collected by centrifugation at $3.465 \times g$ for 10 min/25 °C, washed twice with 500 μL of 50 mM sodium phosphate buffer (pH 7.0), and centrifuged at $13.680 \times g$ for 2 min. The conidia were then stained with 1 mL of 10 μg/mL propidium iodide (PI) for 5 min at 30 °C with incubation at 30 °C in a digital dry bath (D1200; Accu Block, NJ, USA). Lastly, conidia were collected by centrifugation at $13.680 \times g$ for 2 min, washed twice with buffer to remove residual dye, and observed under a light microscope with an epifluorescence system (Eclipse E-200, 400X, Japan).

### Sensitivity to cell wall perturbing agents

Congo red (CR) and calcofluor white (CFW) were evaluated as cell wall stressing agents in terms of the sensitivity of *B. cinerea*. For this assay, the methodology used was from *Plaza et al. (2015)*, with some modifications. Initially, 500 μL of aqueous suspensions of $2.5 \times 10^5$ conidia/mL were pre-incubated in MEB medium with 500 μL of inhibitors ($Cu_{50}$, $Fe_{50}$, and $Cu_{50}$–$Fe_{50}$) for 4 h with agitation at 22 °C. Then, 20 μL aliquots of each preparation were inoculated into growing medium, previously prepared with 2% MEA, and 500 μg/mL each of CR, and CFW in 90 mm diameter Petri dishes. Samples were incubated in darkness

at 22 °C for six days, and mycelial growth diameters were measured daily. All experiments were performed with $n = 9$ samples.

## Pathogenicity test

To assess the pathogenicity of *B. cinerea* in the presence of inhibitors ($Cu_{50}$ and $Fe_{50}$), a test was performed on strawberry fruits (*Fragaria ananassa*) and tomato leaves (*Solanum lycopersicum*). The methodology of *Doehlemann, Berndt & Hahn (2006)* and *Plaza et al. (2015)* was used with some modifications. Before inoculation, the strawberry tissues and detached leaves of tomato were wounded with a pinprick of a 21G syringe and surface-sterilized by immersion with 75% ethanol for 1 min. For inoculation, 5 μL of $2.5 \times 10^5$ conidia/mL suspensions were used with a subsequent incubation step at 25 °C. The lesion diameters were measured using a Caliper up to 4 days after inoculation. Strawberry fruit tests were performed with $n = 15$ samples and $n = 10$ for detached tomato leaves.

## Statistical analysis

The Statistical analysis was carried out according to the protocol of *Leiva-Portilla et al. (2020)*. A one-way analysis of variance (ANOVA) was selected to evaluate significant differences considering *p*- value <0.05 between the means, using Statgraphics Centurion XVI (Statistical Graphics Co., Rockville, MD, USA). Additionally, the least significant difference (LSD) test was used to examine the differences among the media based on a confidence interval of 95%, while the presence of homogeneous groups within each parameter, was validated by the multiple range test (MRT). All the experiments were conducted at least three times.

# RESULTS

## Effect of Cu and Fe on *B. cinerea* mycelial growth on solid media

The antifungal activity of Cu and Fe and their equimolar mixture was assessed against the *B. cinerea* B05.10 strain based on mycelial growth inhibition. For Cu, the concentration range studied was 2–8 mM (Fig. 1A). The results indicated that concentrations greater than 4 mM effectively inhibited the mycelial growth of *B. cinerea* by 85 to 94%, while 2 mM resulted in almost 8% inhibition, after seven days of growth. However, for iron, concentrations between 2 and 20 mM were tested for mycelial growth inhibition (Fig. 1B). At 2–4 mM, no inhibitory effect was observed, since the mycelial growth pattern was similar to the control. In contrast, 6, 8, and 10 mM inhibited the fungus by 6, 29, and 56%, respectively. The results showed that the $IC_{50}$ for iron capable of reducing 50% growth on solid media was equal to 9.08 mM.

Results for the effect of equimolar concentrations of Cu and Fe on the growth of *B. cinerea* B05.10 strain are depicted in Fig. 2. The Cu–Fe mixture at 2 mM showed a low inhibition near the control level, corroborating the findings for Fe and Cu applied individually. An interesting effect was observed for 4 mM of Cu–Fe, which resulted in nearly 75% inhibition of mycelial growth. The toxic concentration for Cu alone resulted in inhibition of 85%, while for iron, the inhibition was not significant. Cu appears to have the predominant inhibitory effect on mycelial growth in this equimolar mixture in comparison

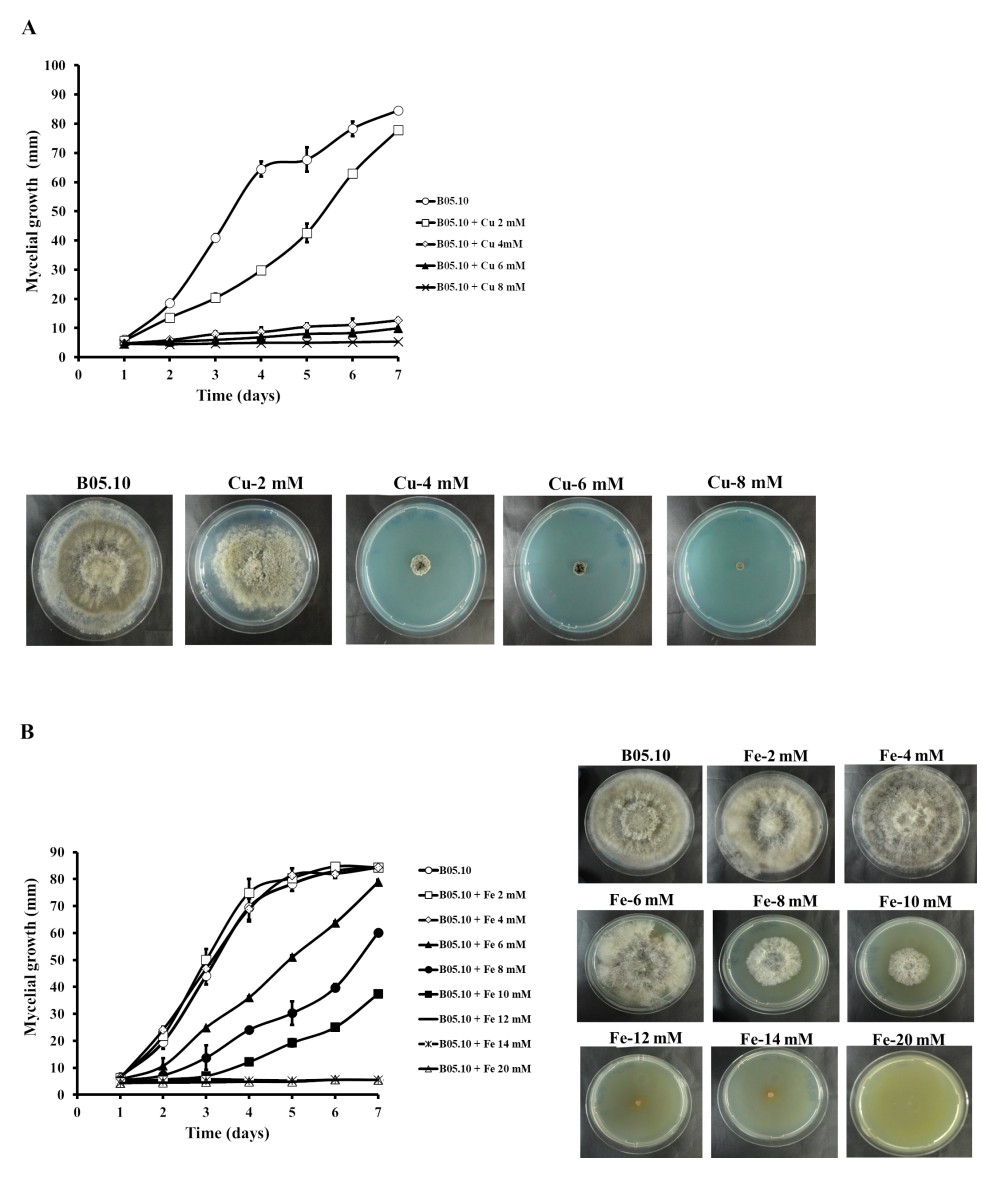

**Figure 1 Effect of metallic inhibitors on the growth of *B. cinerea* in solid media.** Mycelial growth inhibition behavior during seven days of incubation at different concentrations, and the effect on mycelial growth at day 7 of incubation: Cu (A) and Fe (B). Data represent means ± SD from six independent experiments.

to Fe with a minimum contribution (Figs. 2A, 2B). Figure S1 depicts the effect of Cu, Fe, and Cu–Fe on radial growth rate. In the presence of Fe, the radial growth rate was constant in the range 0–4 mM. From 4 mM to 12 mM, there was a continuing decrease until the rate leveled off at 14 mM with a radial growth rate average equal to 0.78 mm/day. In the case of Cu, the decay rate was more drastic from 2 mM until 8 mM, evincing a superior fungitoxicity on B05.10. At the same time, Cu–Fe showed an effect similar to that of Cu alone on radial growth rate from 2 mM, with a subsequent constant decrease until 12 mM, corroborating the predominant effect of Cu in inhibiting mycelial growth in B05.10.

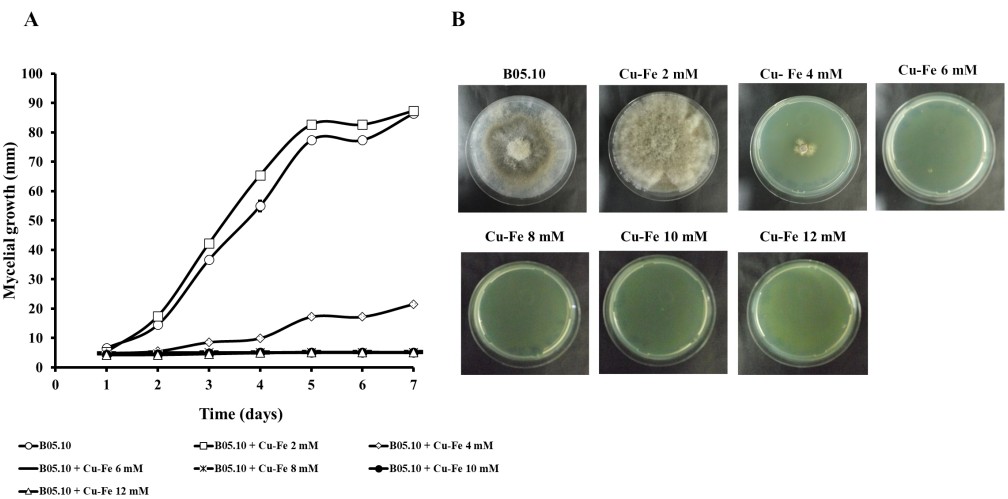

**Figure 2** **Effect of Cu–Fe equimolar mixture on the growth of *B. cinerea* in solid media.** (A) Mycelial growth inhibition behavior during seven days of incubation at different concentrations, and (B) effect on mycelial growth at day 7 of incubation. Data represent means ±SD from six independent experiments.

## Antifungal response of wild strains of *B. cinerea* to treatment with metal inhibitors and metal-fungicides mixtures

To test for possible resistance or different sensitivity patterns, the effects of $Cu_{50}$ and $Fe_{50}$, and the $Cu_{50}$–$Fe_{50}$ mixture were assessed on mycelial growth of the three wild strains: Bc.po03, Bc.vi09 and, Bc.ad03, which have been shown to have different patterns of sensitivity and resistance to specific concentrations of commercial antifungals (Table S3). To determine their identity, all isolates were subjected to DNA sequence analysis (ITS analysis and nuclear protein-coding genes were sequenced) (Table S4), and the procedure was based on a previous work (*Notte et al., 2021*). The partial sequences of the *HSP60* (705 bp), *RPB2* (924 bp), *G3PDH* (690 bp) and *NEP1* (336 bp) genes analyzed by BLAST report a high percentage of identity of all isolates obtained with strains and isolates of *B. cinerea* (Table S5). Therefore, based on these results, the phylogenetic position observed by concatenated genes (*G3PDH, HSP60, RPB2* and *NEP1*) groups all isolates obtained in a single clade together with *B. cinerea* B05.10 (Fig. S2).

Results showed that the wild strains were sensitive to the tested concentrations of $Cu_{50}$, $Fe_{50}$, and $Cu_{50}$–$Fe_{50}$ (Fig. 3). For Bc.po03, isolated from *Porlieria chilensis*, $Cu_{50}$ inhibited 78.05% of the mycelial growth by day 2, compared to control. There was a continuing decrease until day 6, when the metal treatments reached a maximum inhibition of 85.46%. A similar trend with $Cu_{50}$ was observed for the other wild strains, being more marked for Bc.ad03 with a final inhibition of 78.41% (day 6), while Bc.vi09 from *Vitis vinifera* was the most sensitive strain, with a final inhibition of 88.57% (day 6). Wild strains showed greater sensitivity to $Fe_{50}$ than $Cu_{50}$, with a final inhibition greater than 92% in all cases. The inhibitory effect of $Cu_{50}$–$Fe_{50}$ was not significantly different compared to iron alone for Bc.ad03 ($p = 0.05$), Bc.po03 ($p = 0.07$), and Bc.vi09 ($p = 0.73$). These strains were more sensitive to the Cu–Fe mixture than to Cu alone, and we conclude that iron has a

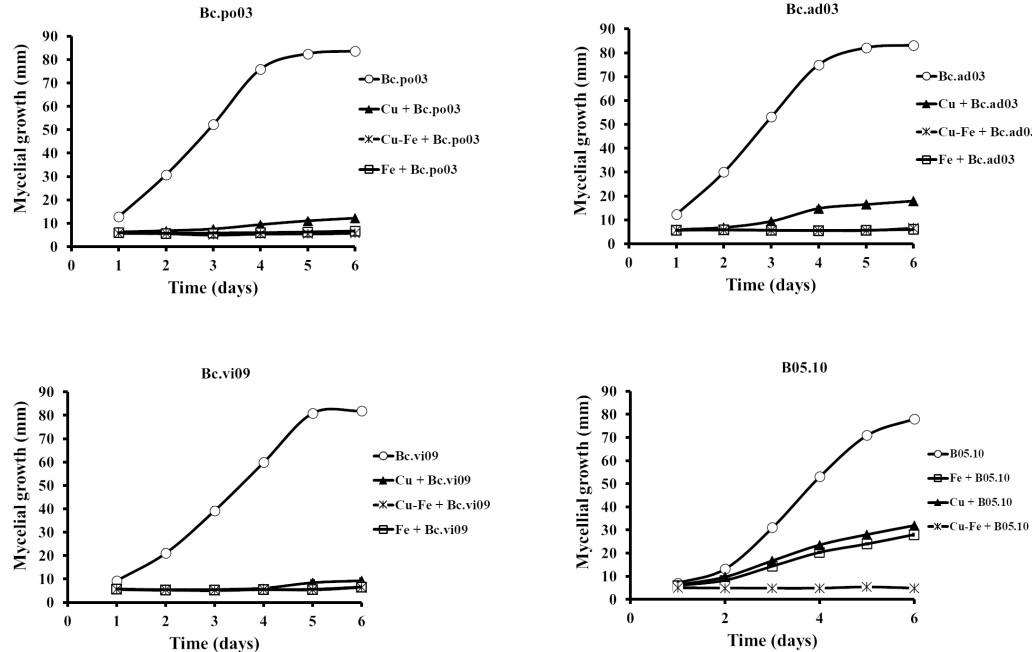

**Figure 3** **Effect of Fe$_{50}$ (9.08 mM), Cu$_{50}$ (2.87 mM), and Cu$_{50}$–Fe$_{50}$ on mycelial growth of different wild strains of *B. cinerea*.** Data represent means ± SD from six independent experiments.

dominant effect over copper. In addition, the wild strains were more sensitive to Cu$_{50}$ and Fe$_{50}$ in comparation to B05.10, however these strains displayed sensitivity similar to the effect caused by Cu$_{50}$-Fe$_{50}$ mixture in B05.10.

Results for the effect on *B. cinerea* regarding mixtures of commercial fungicides-metal inhibitors (Fig. S3), showed that on B05.10 the Cu-fungicides mixture (boscalid and fenexhamid) exhibited a synergistic effect with MI = 1.2 in both cases, but for Cu-iprodione was antagonist (with iM<1); meanwhile on wild strains, the effect was also antagonist predominately. In the case of the mixtures with Fe, an antagonist effect was observed on B05.10 and Bc. vi09, whereas on other wild strains, the depicted effect had a more heterogeneous behavior, only boscalid-Fe on Bc. po03 showed a synergistic effect, while iprodione-Fe on Bc. po03 and Bc.ad03 were the only ones that showed an antagonist effect (Table S6). The observed effects for all the metal-fungicides mixtures could be correlated considering the individual action mode on *B. cinerea* and, the resistance or sensitivity pattern displayed against boscalid, iprodione and fenexhamid by the studied strains.

## Effect of Fe, Cu, and Cu–Fe on conidial germination

The effect on conidial germination of *B. cinerea* in the presence of Cu$_{50}$, Fe$_{50,}$ and Cu$_{50}$–Fe$_{50}$ was also determined. B05.10 without metallic inhibitors (Fig. 4A) revealed a regular pattern of conidial germination over the experimental time period, reaching 44% germination. In the presence of metallic inhibitors, significant inhibition of germination was seen at 4 and 6 h (Fig. 4B). The average percent conidial germination in the presence of Cu or Cu–Fe at

**A**

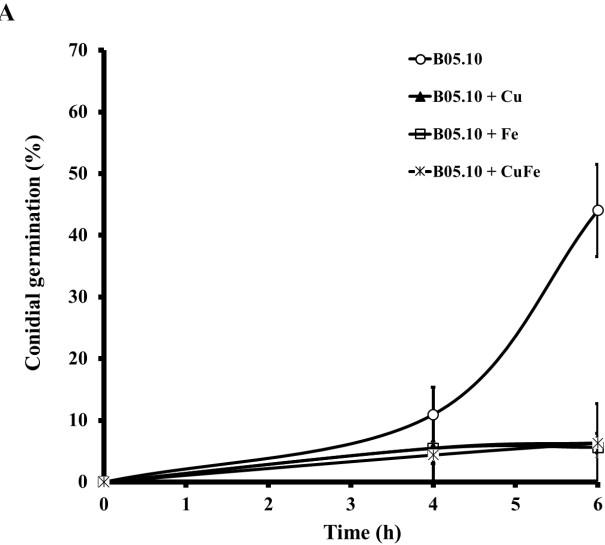

**B**

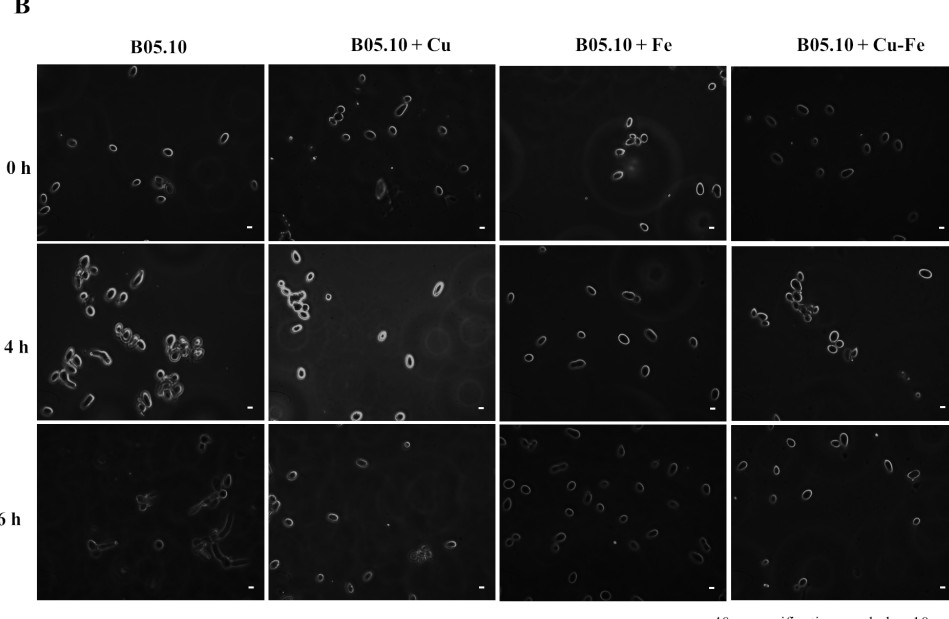

40× magnification, scale bar 10μm

**Figure 4** **Effect of Fe$_{50}$ (9.08 mM), Cu$_{50}$ (2.87 mM), and Cu$_{50}$–Fe$_{50}$ on conidial germination of *B. cinerea*.** (A) Percentage of germination inhibition and (B) images (40 × magnification, scale bar 10 μm) of germinated conidia at different times and metallic inhibitors. Data represent means ± SD from six independent experiments.

6 h was 6.36 ± 1.68 and 6.30 ± 1.60%, respectively, with no significant difference between the two. In contrast, Fe showed an inhibition equal to 5.61 ± 1.61%, and slightly lower than Cu and Cu–Fe.

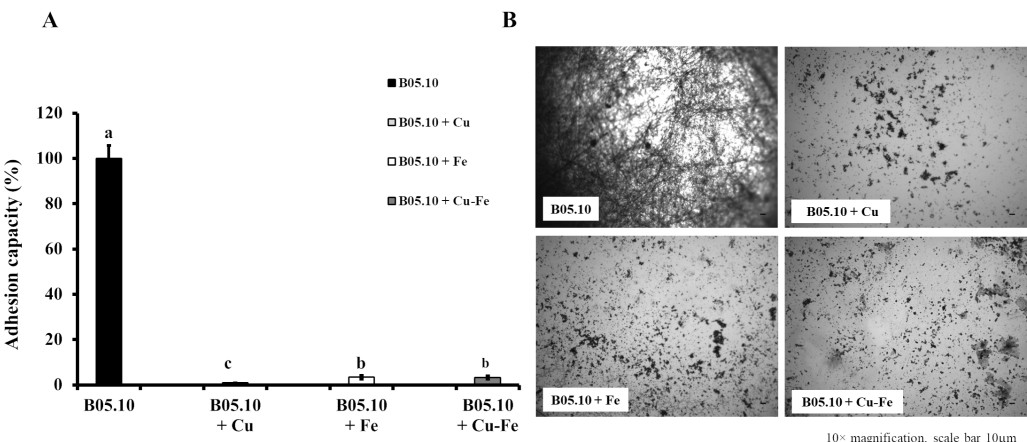

**Figure 5** **Effect of $Cu_{50}$ (2.87 mM), $Fe_{50}$ (9.08 mM), and $Cu_{50}$–$Fe_{50}$ on adhesion capacity of *B. cinerea*.** (A & B) Percent inhibition of metallic inhibitors on the adhesion and biofilm formation, and bound mycelia stained with crystal violet and visualized by fluorescence microscopy (10× magnification, scale bar = 10 μm). All experiments were performed three times with $n = 6$. Different lowercase letters indicate significant differences ($p < 0.05$).

## Effect of Cu, Fe, and Cu–Fe on adhesion capacity of *B. cinerea*

The effect of metallic inhibitors on adhesion capacity was also evaluated (Figs. 5A, 5B). In the presence of $Cu_{50}$, B05.10 showed the lowest adhesion (0.9%) compared to the other inhibitors, with statistically significant differences (Cu *versus* Fe: $p = 0.0208$; Cu *versus* Cu–Fe: $p = 0.0170$, 95%). In the images of stained conidial tissue, remains of conidia with low germination were seen. However, $Fe_{50}$ and $Cu_{50}$–$Fe_{50}$ showed slightly higher adhesion compared to Cu (3%), but this difference was not statistically significant ($p = 0.8509$, 95%).

## Effect of Fe, Cu, and Cu–Fe on cell wall and membrane integrity

Copper and iron are essential micro-nutrients; however, both can also be toxic in cells due to their redox properties and affecting the plasma membrane. The cell wall is a barrier that could help the cell to survive (*Ghaed, Shirazi & Marandi, 2013*; *Tobin, White & Gadd, 1994*). Therefore, the effect of $Fe_{50}$, $Cu_{50}$, and $Cu_{50}$–$Fe_{50}$ on integrity of the cell wall of B05.10 was also analyzed. Conidia treated with metallic inhibitors were examined for their sensitivity to antifungal drugs that specifically affected cell wall assembly or the synthesis of its structural components (Figs. 6 and 7). Congo red affects microfibrils of $\beta$ 1,3-glucans, and calcofluor white alters the assembly of chitin fibrils (*Plaza et al., 2015*; *Torres-Ossandón et al., 2019*). When the effects of $Cu_{50}$, $Fe_{50}$, and $Cu_{50}$–$Fe_{50}$ were compared to the Congo red curves, there was some overlap where minimal alterations and effects on the cell wall were observed (Fig. 6). Some differences in the curves, such as the activity exhibited by Cu on day 5, were not statistically significant ($p = 0.44$, 95%). Evaluating CFW as a disturbing agent (Fig. 7), we found that its effect on B05.10 without inhibitors was lower than the effect of CR without inhibitors. In contrast, CFW treatment with inhibitors revealed that
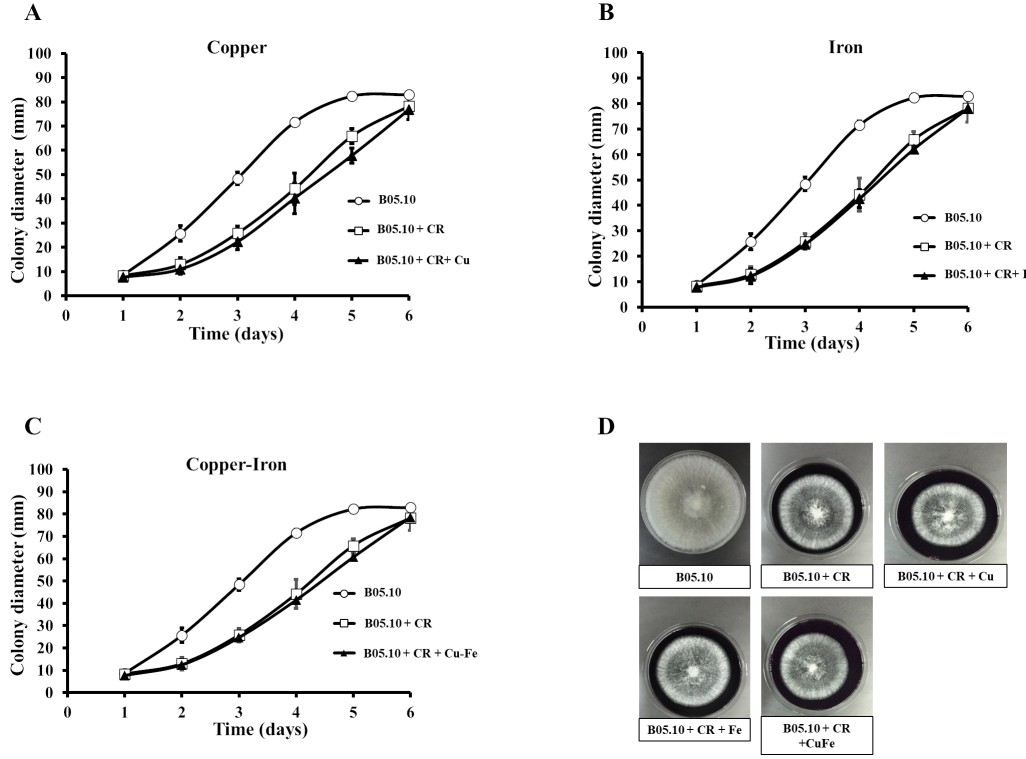

**Figure 6** (A–D) Effect of the metals and Congo red on the cell wall in *B. cinerea*. Sensitivity of B05.10 growth with or without $Fe_{50}$ (9.08 mM), $Cu_{50}$ (2.87 mM), and $Cu_{50}-Fe_{50}$ in solid media with Congo red (CR). Data represent means $\pm$ SD from nine independent experiments.

the effect on the cell wall was not due to the metals, which overlapped, but only with non-significant statistical differences.

The effects of $Cu_{50}$, $Fe_{50}$, and $Cu_{50}-Fe_{50}$ on membrane integrity of *B. cinerea* were also studied using PI as a fluorescent indicator of *cell* viability. Untreated conidia and conidia treated with inhibitors for various times were stained with PI and imaged by fluorescence microscopy (Figs. 8A, 8B). The effect of Fe and Cu on the membrane ranged from 2 to 6% at 0 h. The Cu–Fe combination exerted an effect close to 20%. Effects on the membrane were more significant as time progressed to 6 h, where conidia treated with Cu, Fe, and Cu–Fe showed damage to membranes in 20.5, 15.51, and 30.85% of samples, respectively. The results suggested that the mixture of these metals possibly exerted a synergistic effect in damaging the cell membranes of *B. cinerea*, exhibiting a higher number of nonviable conidia.

## Pathogenicity test

The pathogenicity patterns of B05.10 in the presence of $Fe_{50}$ and $Cu_{50}$ were also assessed as performed on strawberry fruits and tomato leaves (Fig. 9). The growth of *B. cinerea* in the fruits was measured on days 0 and 4 after inoculation. The results suggest that metal inhibitors were able to inhibit mycelia growth after inoculation in wounds on the fruits, indicating the antifungal effect at 2.87 mM and 9.08 mM for copper and iron, respectively.

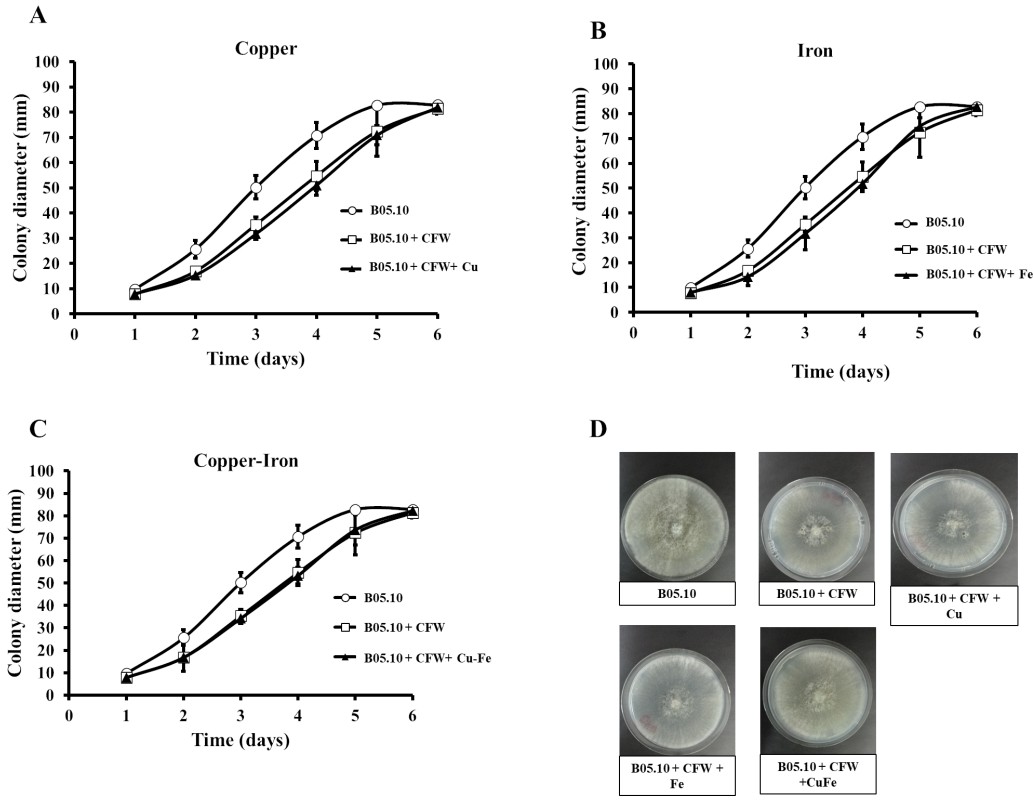

**Figure 7** (A–D) **Effect of the metals and Calcofluor white on the cell wall in *B. cinerea*.** Sensitivity of B05.10 with or without Fe$_{50}$ (9.08 mM), Cu$_{50}$ (2.87 mM), and Cu$_{50}$–Fe$_{50}$ grown on solid media with calcofluor white (CFW). Data represent means ± SD from nine independent experiments.

Regarding fruit senescence, the comparison of fruit appearance from day 0 to 4 showed only minor changes with copper compared to strawberries inoculated in the presence of iron. Regarding the test performed on tomato leaves with metal inhibitors, after 4 days of incubation iron was able to inhibit mycelial growth by 53.8% compared to B05.10, while copper inhibited to a lesser extent with 25.9%.

# DISCUSSION

In this work, the results demonstrate that Cu, Fe, and Cu–Fe exhibit antifungal activity against *B. cinerea* B05.10, grown on solid and liquid media. The effect of copper on mycelial growth showed that a concentration of 2 mM was minimally toxic to the B05.10 strain. This can be explained because the Cu requirements of microorganisms are usually satisfied with low concentrations of this metal, on the order of l-10 µM, which serves as a micronutrient for the growth and proliferation of filamentous fungi (*Cervantes & Gutierrez-Corona, 1994*). A similar effect was observed by *Buddhika, Savocchia & Steel (2020)* who showed that Cu$^{2+}$ stimulated growth of *B. cinerea* at concentrations below 0.6 mM. At concentrations greater than 2 mM, especially between 4–8 mM, copper inhibited the mycelial growth of *B. cinerea*, with an IC$_{50}$ of 2.87 mM. This value was comparable to those reported by *Judet-Correia et*

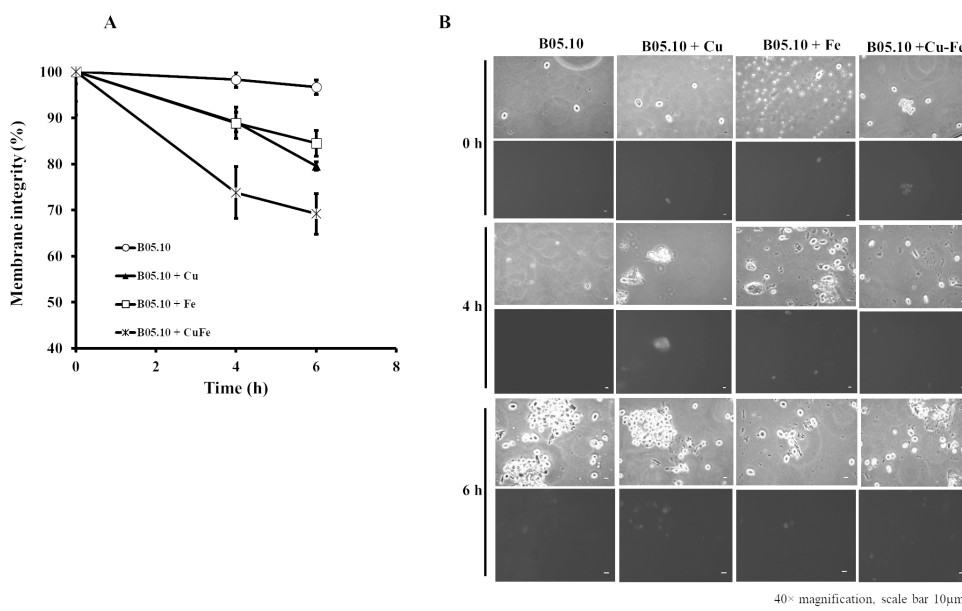

**Figure 8 Effect of the metals on plasma membrane integrity of _B. cinerea_.** (A) Effect of $Fe_{50}$ (9.08 mM), $Cu_{50}$ (2.87 mM), and $Cu_{50}-Fe_{50}$ on membrane integrity of _B. cinerea_. (B) Effect on _B. cinerea_ cells treated with PI (40 × magnification, scale bar = 10 $\mu$m). Data represent means ± SD from six independent experiments.

_al. (2011)_ on two isolated strains of _B. cinerea_ (BC1 and BC2) with $IC_{50}$ values equal to 2.60 and 2.21 mM, respectively. It has been reported that Cu can be highly toxic to microbial cells when present at high levels in its free ionic form, $Cu^{2+}$. This effect is mainly related to its interaction with nucleic acids, the oxidation of membrane components, and the modification of enzyme active sites (_Cervantes & Gutierrez-Corona, 1994_). Accordingly, these processes can be linked to the capacity of copper to produce toxic hydroxyl free radicals, generating redox imbalance at the cellular level. These effects could explain the results found in our study where high concentrations of copper affected mycelium growth in this fungus (_Cervantes & Gutierrez-Corona, 1994_; _Lamichhane et al., 2018_). In addition, laccases are inducible enzymes produced by filamentous fungi during the colonization of host plant tissues, and their activity is inhibited by excess Cu, which prevents the infection (_Schouten et al., 2002_; _Buddhika, Savocchia & Steel, 2020_; _Hernández-Monjaraz et al., 2018_). Other authors showed evidence that copper toxicity in fungi could occur because copper displaced essential metal cofactors from their native protein binding sites or blocked ligand interactions (_Borkow & Gabbay, 2005_).

With regard to iron, it is an essential element for most organisms and a critical factor for fungal virulence in pathogenic species (_Gerwien et al., 2018_; _Liu et al., 2020_). Its role is important as a vital component of iron-sulfur clusters required for activation of nuclear proteins involved in DNA repair (_Robinson, Isikhuemhen & Anike, 2021_). Our data show that low iron concentrations in the range of 2–4 mM did not affect the growth of _B. cinerea_. However, iron concentrations greater than 4 mM progressively inhibited the fungus, reaching a value of 56% for 10 mM, with inhibition percentages near 93% for 12–20 mM.

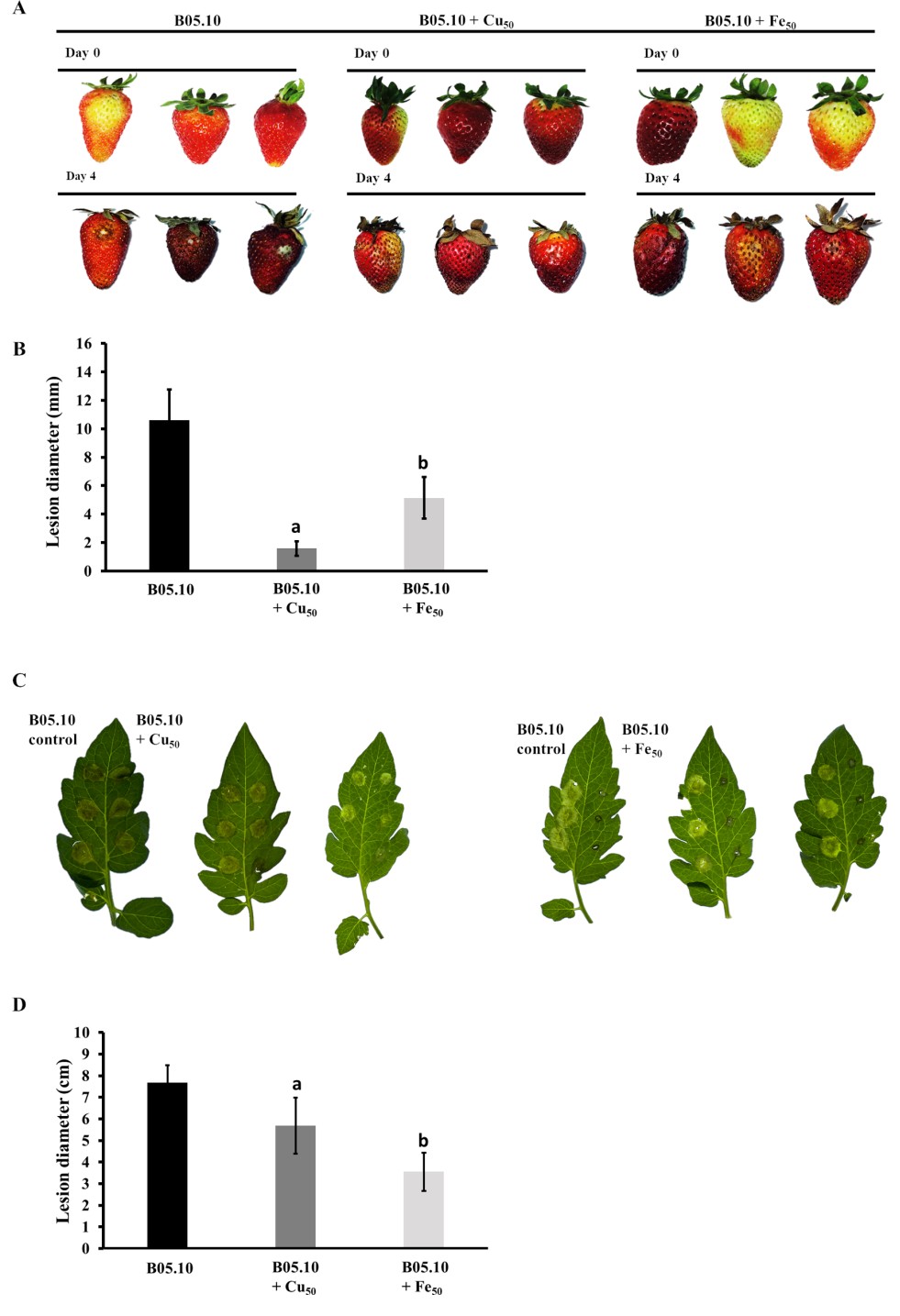

**Figure 9** **Pathogenicity assay of the *Botrytis* B05.10 strain in presence of metallic inhibitors.** (A & B) Pathogenicity test carried out for B05.10 ($2.5 \times 10^5$ conidia/mL) in the presence of $Fe_{50}$ (9.08 mM), and $Cu_{50}$ (2.87 mM) placed inside wounded strawberry fruits, with comparisons at days 0 and 4. Data represent means $\pm$ SD from 15 independent experiments. (C & D) Droplets of 5 ml containing $2.5 \times 10^5$ conidia/ml for B05.10 in the presence of $Fe_{50}$ (9.08 mM), and $Cu_{50}$ (2.87 mM) placed onto the surface of a tomato leaf. Lesion development was scored after four dpi. Data represent means $\pm$ SD from 10 independent experiments.

The inhibition pattern obtained for iron, was compared with data from *Fleurat-Lessard et al. (2011)* for *B. cinerea* (strains Bc 112 T, Bc 162 V, Bc 344 T, Bc 916 T), showing that the mycelial growth was not significantly affected by 1 mM, but there was a strong effect at 10 mM and complete inhibition at 20 mM. A $IC_{50}$ value for iron of 9.08 mM was determined, a value higher than the $IC_{50}$ of Cu. This result could be explained by one of the most plausible mechanisms, where fungitoxicity has been associated with redox cycling between the ferric and ferrous forms, catalyzing the generation of dangerous free radicals that are toxic to plants and pathogen cells (*Liu et al., 2020*). Iron homeostasis is therefore tightly regulated (*Greenshields, Liu & Wei, 2007*). To maintain Fe homeostasis, fungi have to balance Fe acquisition, storage, and utilization to guarantee an adequate supply and avoid a toxic excess (*Misslinger et al., 2021*). Preserving the suitable balance of Fe between deficiency and toxicity requires fine-tuned control of iron uptake and storage (*Haas, Eisendle & Turgeon, 2008*). According to *Gerwien et al. (2018)*, several pathogens have evolved elaborate systems to acquire iron from their environment and to store the excess at a high concentration in a nontoxic form for times of need. This could partly explain the tolerance observed for iron at concentrations between 4 and 6 mM in our experiments.

A combination treatment of equimolar concentrations of Cu and Fe was also assessed on the *B. cinerea* B05.10 strain. Significant antifungal activity of Cu–Fe at 4 mM or higher was evidenced in comparison to the individual effects Cu or Fe. Interestingly, the *B. cinerea* cells exhibited a high sensitivity to low concentrations of Cu as described before, but a possible synergistic effect with Fe at concentrations greater than 4 mM. Our results are consistent with other authors who have established that copper is highly toxic towards various microorganisms including bacteria, yeasts, *Plasmopara viticola,* and *B. cinerea* (*Judet-Correia et al., 2011*; *Zoroddu et al., 1996*). According to *Gerwien et al. (2018)*, several common biological metals have similar divalent cation properties in binding ligands but different catalytic functions. For example, host-induced metal excess and oxidative stress can result in 'mismetallation', replacing the normal metal cofactor of an enzyme with a different metal. This process inhibits the role of microbial enzymes that require specific metals as cofactors (*Imlay, 2014*). In the case of copper, some of its toxic effects arise from the high capacity of $Cu^+$ under anaerobic reducing conditions, to displace other metals from their coordination sites, while, $Cu^{2+}$ forms the most stable complexes of the divalent transition metals (*Gerwien et al., 2018*; *Irving & Williams, 1948*). Similar to copper, iron has a strong propensity for replacing other metals in reactive enzymatic centers, which generally disrupts the enzyme function (*Martin & Imlay, 2011*; *Vance & Miller, 1998*), generating an imbalance in the homeostasis of these metals. This could explain the apparent synergistic effect between Cu and Fe on *B. cinerea*.

The effect of Cu, Fe, and Cu–Fe on mycelial growth *B. cinerea* B05.10 strain was additionally compared with the effect of this pathogen on wild strains, showing that these strains were more sensitive to the test concentrations of metallic inhibitors than B05.10. At the same time, with the mixture of metallic inhibitors, wild strains displayed sensitivity similar to the effect caused by iron, confirming the dominant effect of this metal. A similar antifungal pattern was reported by *Notte et al. (2021)* for the diverse phenotypes and genotypes of the *B. cinerea* isolates studied. It is important also to consider that diverse

*B. cinerea* isolates displayed different degrees of virulence and response to antifungal agents, which could be related to intraspecific differentiation and different degrees of host adaptation of this pathogen (*Plesken et al., 2021*). In addition, it has been reported that, some genes in *B. cinerea* that are important for virulence, may contribute differently depending on the genetic background (*Pinedo et al., 2009*; *Siewers et al., 2005*).

On the other hand, when metal-fungicides were evaluated, the synergistic effect observed on B05.10 by the Cu-antifungal mixture could be associated with the disruptive effect exerted by copper on the cell membrane in *B. cinerea* and, the mode of action of the fungicides. For example, in the case of fenexhamid and boscalid, it has been reported that they interfere with the biosynthesis of ergosterol (*Debieu et al., 2013*; *Pappas & Fisher, 1979*),  a key component in the architecture of the cell membrane, affecting its production and therefore the growth of the fungus. While, in the case of boscalid, it is a succinate dehydrogenase inhibitor, affecting the proper functioning of cellular metabolism (*Cherrad et al., 2018*; *Piqueras, Latorre & Torres, 2014*). Then, possibly, in these mixtures, once the copper damages the membrane, the antifungals can easily enter the cell and exert their inhibitory effect. Regarding mixtures with Fe on B05.10, the predominant effect was antagonistic, resulting in a decrease of the known individual efficacy of each inhibitor, an opposite result than obtained for Cu-boscalid and Cu–fenhexamid. Then, according to our results of membrane integrity, even though Fe showed a similar effect to Cu on the cell membrane, a close result was expected, but in this case other additional interactions could be possible modifying the inhibitory effect, enhancing the antagonistic effect. Concerning wild strains, the predominant antagonistic effect with metal-fungicides mixture could be correlated with the previous sensitivity and resistance pattern studied against iprodione, boscalid and fenexhamid, and the cellular infrastructure or intraspecific differentiation developed in the evolution of these wild strains and their genetic background. However, several additional studies still are necessary to understand the interactions involved in this scenario.

Conidial germination of the B05.10 *B. cinerea* strain was likewise affected by the presence of Cu, Fe, and Cu–Fe in liquid medium. For $Cu_{50}$ and $Cu_{50}$–$Fe_{50}$ at 6 h, the inhibition of conidial germination did not show significant differences ($p = 0.9675$, 95%), while $Fe_{50}$ had an inhibitory effect slightly below them ($p = 0.5845$ (Fe *vs.* Cu); $p = 0.6313$ (Fe *vs.* Cu–Fe)). According to *Judet-Correia et al. (2011)*, copper may bind to the surface of spores during germination. Thus, a period of time is required for the detoxification process and selection of surviving spores. The same authors reported that copper concentrations from 0–4 mM did not affect the lag time for the growth of *B. cinerea*, suggesting that the germination did not depend on these copper concentrations. In contrast, copper delayed the germination of spores at concentrations >4 mM, needing more time to detoxify the medium by binding Cu at the surface of some spores. In our case, the $IC_{50}$ concentration of Cu was 2.78 mM. The toxic effects of Cu on the growth of other fungi have also been reported (*Jaworska et al., 1996*; *Tkaczuk, 2005*), indicating that this phenomenon could be dependent on the fungal species and physicochemical factors, as well as metal concentration (*Tobin, White & Gadd, 1994*). Regarding the effect of iron, a relevant inhibition of conidial germination was observed at 4 and 6 h, with a concentration of 9.08 mM for $Fe_{50}$ being calculated in

this case. $Fe^{2+}$ has been reported within a group of other tested cations ($Ca^{2+}$, $Mg^{2+}$, $K^+$) where concentrations in the range of 0.001–1 mM did not influence conidial germination (*Nassr & Barakat, 2016*). However, concentrations greater than 10 mM resulted in an abrupt decrease in germination, especially for $Fe^{2+}$. It is likely that, before germination, conidia are not altered at low cation concentrations in the growth substrate. Germ tube growth becomes more sensitive to a wide range of cations ($Ca^{2+}$, $Mg^{2+}$, $K^+$, and $Fe^{2+}$). In addition, $Fe^{2+}$ seems to provide an important nutritional source for germ tube growth at low concentrations (0.001 M), and it was found to support germ tube growth and elongation (*Nassr & Barakat, 2016*). Lastly, the combination of $Cu_{50}$–$Fe_{50}$ showed a toxic behavior on B05.10 germination dominated mainly by the effect of copper in the mixture.

Conidial germination was correlated with adhesion capacity for *B. cinerea* in the presence of metallic inhibitors, which allowed us to evaluate this pathogen's ability to adhere to environmental hosts (fruits, leaves, *etc.*), to advance in its proliferation and infection process. The presence of copper in the medium resulted in the lowest adhesion capacity of the strain. Iron and the mixture Cu–Fe showed an adhesion capacity 3.3-fold times higher than Cu, but without statistically significant differences. This result could be attributable to Fe in the Cu–Fe mixture. In the fungal adhesion process, biofilm formation was considered an important virulence attribute of these pathogens (*Fanning & Mitchell, 2012*). Specifically, for *B. cinerea*, adhesion was strongly related to the synthesis of glycoproteins in the outermost layer of the cell wall, which act as 'stickers' on the host, facilitating propagation on the surface (*Plaza et al., 2015*). Thus, a possible mechanism or effect of metallic inhibitors on adhesion could be related to an inhibition of the synthesis of this type of protein. It is essential to consider the specific mechanism of the fungal-host attachment process. Adhesion occurs at multiple stages of fungal morphogenesis, where adhesion can be related to zoospores, conidia, germlings, hyphal aggregates, and infection cushions (*Epstein & Nicholson, 1997*). Two specific stages have been established in the adhesion of conidia and germlings in *B. cinerea*. The first stage occurred immediately upon hydration of conidia, and was characterized by hydrophobic interaction forces. The second stage of adhesion occurred after viable conidia had been incubated for several hours under conditions that promoted germination. These asseverations were in line with the reported correlation between adhesion and germination observed in *B. cinerea* (*Doss et al., 1995*; *Torres-Ossandón et al., 2019*). The inhibitory effect on the germination of B05.10 shown by Fe, Cu, and Cu–Fe can provide a key for understanding the actions observed in this study on the adhesion capacity, where non-germinated conidia remained attached to the surface, as observed after the aspiration process.

Cell membrane damage can result in loss of diffusible cellular solutes, oxidation of membrane components, and increased permeability of the cell to external material (*Cervantes & Gutierrez-Corona, 1994*; *Tobin, White & Gadd, 1994*). Our results suggest that metallic inhibitors can affect the membrane integrity of B05.10. In this strain, when treated with $Cu_{50}$, $Fe_{50}$, or $Cu_{50}$–$Fe_{50}$, we observed a higher number of stained conidia in assays with PI. Comparing the effect exerted by iron and copper on membrane integrity, a more significant number of damaged conidia was observed with the individual presence of these metals. Exposure to high Cu concentrations induced a disruption in membrane integrity,

producing a selective change in plasma membrane permeability, leading to loss of cell viability (*Borkow & Gabbay, 2005*; *Fleurat-Lessard et al., 2011*). The equimolar mixture of metals exhibited a possible synergistic effect on damage to the cell membranes of *B. cinerea*, compared to the effects of metal inhibitors individually. Several approximations related to the copper fungitoxicity indicated that this metal could damage membranes through various mechanisms, among which are redox imbalance at the cellular level with damage to the membrane, delays in the germination of its spores, and inhibitory effects on laccase activity, preventing infection (*Buddhika, Savocchia & Steel, 2020*; *Judet-Correia et al., 2011*; *Lamichhane et al., 2018*). In the case of Fe, the total elucidation of its antifungal mechanism has not been accomplished. It has been associated with an attack on pathogens by induction of reactive oxygen species at the cellular level (*Fleurat-Lessard et al., 2011*). In biological systems, where both copper and iron coexist, they can interact with oxygen in their free unbound form by catalyzing the Fenton/Haber-Weiss reaction, which increases ROS production. If cellular antioxidants cannot adequately counterbalance this production, the oxidative damage of proteins, lipids and nucleic acids ensues (*Letelier et al., 2010*; *Sharma et al., 2012*).

In fungi, the cell walls are fundamental for viability and pathogenicity, determining fungal morphology, providing osmotic protection, and regulating material exchange with the environment, adhesion to a substrate, or penetration (*Cantu et al., 2009*). Structurally, fungi cell walls are composed of linear polymers of chitin, $\alpha$-1,3; $\beta$-1,3- and $\beta$-1,6-glucans, and mannoproteins (*Adams, 2004*). As has been shown, CFW preferentially binds to chitin in fungi and interferes with cell wall assembly, while CR binds to $\beta$1,3-glucan and alters the assembly of microfibrils at the cell wall (*Ram & Klis, 2006*). The wall is the first cellular site of interaction with metal species and appears to have a major role in biosorption due to possession of numerous uptake sites (*Ghaed, Shirazi & Marandi, 2013*; *Tobin, White & Gadd, 1994*). Our data suggest that in the presence of CR or CFW plus metallic inhibitors ($Cu_{50}$, $Fe_{50}$, and $Cu_{50}$–$Fe_{50}$), *B. cinerea* showed non-statistically significant differences compared to the control strain without metal treatment, indicating no major change in cell wall integrity. Chitin and chitosan are significant metal-absorbing substances in cell walls of fungi (*Ghaed, Shirazi & Marandi, 2013*; *Tobin, White & Gadd, 1994*; *Volesky, 1990*). Our data suggest that metal could be present in the cell wall but would not alter the chitin and glucan contained in *B. cinerea*. However, additional studies are required to corroborate these findings. For example, future studies can use HPLC quantification of the $\beta$-glucans and chitin to compare the synthesis of polysaccharides with and without metallic inhibitors.

Lastly, the study of the pathogenicity of *B. cinerea* in the presence of the metallic inhibitors showed that copper and iron also affected virulence of *B. cinerea* on strawberries and tomato leaves. Specifically, the strawberries senescence was observed to be lower for fruits inoculated with copper compared to iron. However, in a host as tomato leaves the inhibition pattern was the inverse of that observed in strawberry fruits, where iron inhibited *B. cinerea* more than copper did. These differences can be analyzed from different points. Although copper and iron are essential, an excess could result in the generation of ROS through the Fenton/Haber-Weiss reaction and inactivation of enzymes by replacement of normal metal cofactors, thus affecting the virulence mechanism (*Gerwien et al., 2018*).

This could be associated with defects in germination and membrane damage caused by copper and iron in these fungi as explained above. Additionally, among the virulence factors, the ambient pH is considered the most important physiochemical factor affecting cell growth, development, and the interaction host-*B. cinerea* (*Rascle et al., 2018*). Hosts as the fruits typically exhibiting a pH ranging from 3.32 to 4.39, whereas leaves, stems, and roots exhibit a higher pH ranging from 5.81 to 6.3 (*Hua et al., 2018*). Precisely this case involves the typically acidic pH of fruits, where these metals are more soluble, resulting in pH-dependent systems of metal homeostasis (*Gerwien et al., 2018*). This suggests that pH may have played a role in the pathogenicity however, more studies devoted to the analysis of genes involved in iron and copper metabolism in *B. cinerea* are required, considering the differences in inhibition patterns observed depending on the type of host using the same metal concentration.

## CONCLUSIONS

New insights are presented in this work for characterizing the inhibitory effect of Cu and Fe, separately and together, on *B. cinerea*. Our data shows that Cu and Fe can successfully inhibit the mycelial growth and conidial germination of *B. cinerea*, while tests of the Cu–Fe combination revealed that the inhibitory effect of Cu was dominant over Fe. The adhesion capacity of *B. cinerea* in the presence of the metallic inhibitors was correlated with the effect observed on conidial germination, where inhibited conidia continued to adhere to the surface. A negative effect exerted by metallic inhibitors on the plasma membrane integrity of *B. cinerea* was also observed. Our findings revealed that the cell wall was not altered by the antifungal effects of Cu, Fe, and Cu–Fe. However, future studies are required to corroborate these findings. Lastly, the pathogenicity test showed that the virulence level was controlled by the individual presence of Cu and Fe. The broad characterization developed in our work represents new contributions about the inhibitory effect of these metals on *B. cinerea* that can increase understanding of the mechanism involved at the cellular level and promote development of new metal-based fungicides to ensure sustainability of agri-food crops.

## ACKNOWLEDGEMENTS

We thank Dr(c) Matias Poblete for help with Fig. S2 and MSc. Bárbara Marambio-Alvarado and Lila Olivares-Urbina for their assistance in work at the Laboratory of Biochemistry and Molecular Biology, Department of Biology, University of La Serena, Chile.

### Funding

The authors received funding from the "Dirección de Investigación y Desarrollo (DIDULS-TTP212128 and TTP222122)" and the Doctoral Program from the University of La Serena-Chile. Vilbett Briones received support from the ANID Fondecyt Regular project

(2022–2024), Grant No. 1220845 for the APC. The funders had no role in study design, data collection and analysis, decision to publish, or preparation of the manuscript.

## Grant Disclosures

The following grant information was disclosed by the authors:
Dirección de Investigación y Desarrollo: DIDULS- TTP212128,  TTP222122.
Doctoral Program from the University of La Serena-Chile.
ANID Fondecyt Regular:  1220845.

## Competing Interests

The authors declare there are no competing interests.

## Author Contributions

- Fátima Rodríguez-Ramos conceived and designed the experiments, performed the experiments, analyzed the data, prepared figures and/or tables, authored or reviewed drafts of the article, and approved the final draft.
- Vilbett Briones-Labarca conceived and designed the experiments, analyzed the data, authored or reviewed drafts of the article, and approved the final draft.
- Verónica Plaza conceived and designed the experiments, performed the experiments, analyzed the data, prepared figures and/or tables, authored or reviewed drafts of the article, and approved the final draft.
- Luis Castillo conceived and designed the experiments, analyzed the data, authored or reviewed drafts of the article, and approved the final draft.

## DNA Deposition

The following information was supplied regarding the deposition of DNA sequences:
The sequences are available at GenBank:
Bc.ad03 ITS DNA: OP852422
Bc.po03 ITS DNA: OP852423
Bc.vi09 ITS DNA: OP852424
Bc.ad03 partial sequence RPB2 gene: OP859131
Bc.po03 partial sequence RPB2 gene: OP859132
Bc.vi09 partial sequence RPB2 gene: OP859133
Bc.ad03 partial sequence HSP60 gene: OP859134
Bc.po03 partial sequence HSP60 gene: OP859135
Bc.vi09 partial sequence HSP60 gene: OP859136
Bc.ad03 partial sequence G3PDH gene: OP859137
Bc.po03 partial sequence G3PDH gene: OP859138
Bc.vi09 partial sequence G3PDH gene: OP859139
Bc.ad03 partial sequence NET1 gene: OP859140
Bc.po03 partial sequence NEP1 gene: OP859141
Bc.vi09 partial sequence NEP1 gene: OP859142

## Data Availability

The raw data are available in the Supplemental Files.

## Supplemental Information

Supplemental information for this article can be found online at http://dx.doi.org/10.7717/peerj.15994#supplemental-information.

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
