# Peer review of "Iron and copper on Botrytis cinerea: new inputs in the cellular characterization of their inhibitory effect"

_PeerJ, doi:10.7717/peerj.15994_

## Round 0.1 · original submission · Major Revisions

The manuscript was assessed by two experts in this field who think that might be suitable for publication after addressing their concerns. Most of them are related to manuscript preparation and English usage, which I think are easy to address. There are two experiments suggested by Reviewer 2 that I consider relevant for manuscript improvement. Quantitative data in virulence assays (it is mentioned they were collected but not shown in the manuscript) and the combination of these elements with fungicides to assess any synergistic effect.

Reviewer 1 ·

Basic reporting

In general, the manuscript is structured correctly. It is clearly written. The raw data is displayed.

Experimental design

The experiments carried out are adequate to answer the initial research questions. The procedures and methods are understandable and sufficiently described to make reproductions if necessary.

Validity of the findings

The conclusions of the work are supported by the results shown.

Additional comments

The following corrections and suggestions are for the purpose of improving the manuscript.

line 141: Define this gene G3PDH, HSP60, RPB2 and NEP1
line 204: Define MEB
line 211: The same for OD.
line 220: I suggest changing rpm to x g
lines 235, 236, 239: ibidem
line 386: Italic font
line 549: ibidem

The solubility of metals in the medium and their bioavailability is closely related to the pH; therefore, I suggest you place the pH of the solutions and media used in the experiments.

Reviewer 2 ·

Basic reporting

In general terms, the MS is understandable and easy to follow. However, English requires significant improvements, some of which (not all) I will indicate later. In general terms, the MS requires professional English editing to be read and understood better. Some references concerning copper and iron in B. cinerea are absent, and more depth is needed in the Introduction. The discussion is too extensive and, at the same time, with a depth that could be significantly improved.

I also suggest being more precise about the experiments and background that lead to evaluating the membrane and wall effects of metals.

Experimental design

No comment

Validity of the findings

A little more work should be done on the rationale behind each experiment. My main concern is related to the differences in virulence, which, as they are presented today, are qualitative but not quantitative.

Additional comments

Since I routinely follow the Botrytis literature, I was immediately struck by the second citation mentioned in line 42 (year 2020). Interestingly, in the cited paper, the primary intent is not to describe Botrytis as the causal agent of a major disease in agricultural fields (as it is in Dean et al., 2002); quite the contrary, the paper describes a mutant of an iron transporter system in B. cinerea (RIA system). Is iron essential for Botrytis virulence? What is known about it? Since it is one of the metals discussed in this study, it seems essential that the authors contextualize such research much better. In fact, it is not addressed concerning the iron findings reported in the manuscript. Furthermore, there is minimal introduction regarding what is known about siderophores in Botrytis. Regarding the RIA system, Condon et al 2014 is correctly cited, but nothing is mentioned regarding the paper cited in line 42. Regarding copper, the situation is similar. There needs to be more information regarding capture and chelation mechanisms (some even described in fungi for the first time) or copper detoxification mechanisms.

The materials and methods section also resembles the wording used in a thesis rather than a manuscript. There are too many details and, conversely, too little information on what you are seeking to accomplish. I also need clarification on why there are two sections for determining mycelial growth as a function of the presence of metals but one individual section for strain B0510, and another for the isolates. Many things can be summarized and rearranged.

Finally, regarding the virulence assays, it is mentioned that measurements of the lesions were made, but this information is not contained in the document. The related figure only shows the fruits, but there is no quantitative data on the lesions observed.

In summary, more detailed work is required in the introduction and discussion. It would also be optimal to have infection assays in another plant model where it is easier/simpler to determine lesion sizes. And finally, considering that the paper generally emphasizes the importance of identifying new Botrytis control tools, most of which are chemical, it would be interesting for the authors to evaluate what happens when Cu and/or Fe are mixed with fungicides, in particular, if the authors take advantage of isolates that are resistant to a fungicide. What happens in this case? Can the resistant isolates become sensitive if Cu and/or Fe are added in tiny amounts? This result can greatly improve the discussion of the results.

Minor comments:

Line 46: replace “parts” with “organs”.
Line 67: …metals must be maintained within an acceptable CONCENTRATION RANGE?
Line 96: P. expansum is mentioned for the first time in the manuscript.
Line 107: Please include (cite) the various studies indicated in the text.
Line 142: Do you mean pairs, no combination?
Line 138-139: please, double-check citation style in the indicated lines and throughout the M&M section. (e.g., 141-142)
Line 283: …almost 8% inhibition AFTER 7 DAYS OF GROWTH?
Line 288: …the effect was similar to the control…DO YOU MEAN NO EFFECT WAS OBSERVED?
Line 288-289: Please remove the following sentence as it is a “big” conclusion for a simple experiment. “…corroborating the role of iron at specific concentrations as a nutrient for the growth and development of this pathogen”.
Line 320-321: these isolates, not this isolate.
Line 324: BLAST, not Blast. Also, remove NCBI. Please, indicate the % of identity as well.
Line 373: Figure 7, please arrange the legend of each plot as in Figure 6.
Line 386: italics on B. cinerea. (Also in line 549).
Line 415: B05.10, not B.0510
Line 518: please, be careful: English needs to be revised.
Line 542: Replace the second “B05.10” with “This strain, when treated with…”

---

## Round 0.2 · Major Revisions

After a second revision by the original Reviewers, one of them still thinks the manuscript content should be improved. After reading the explanations given for the requested additional experiments, I think these elements are essential to improve the content of this manuscript and should be included in a revised version of the text.

Reviewer 1 ·

Basic reporting

No comment.

Experimental design

No comment.

Validity of the findings

No comment.

Additional comments

In general, the aforementioned corrections have been made and suggestions for improving the manuscript have been taken into account. Just a small correction though, check the centrifugal force units again, it's not g, it's xg.

From here on, there is no other comment from me.

Reviewer 2 ·

Basic reporting

In general terms, the manuscript still requires substantive improvements in form (incomplete figures such as Figure 1B that lacks the Fe-20 mM photo; figures that are not fully polished), in substance because there are essential concerns with the literature, and some interpretations made in the discussion. For example, I am afraid the sentence in lines 93-96 needs to be corrected: the co-existence of RIA and siderophores production and/or incorporation was reported several years ago, and certainly not in B. cinerea. In this regard, I still think the authors need to review the (fungal) iron-related literature more deeply to better introduce the field in this manuscript. All combinations in fungi do exist: RIA, RIA+siderophores, and siderophores-only.

In the first round of revision, the authors were asked to perform additional experiments. These are very easy and simple further experiments. However this was strongly suggested, given that the authors employed three isolates with different sensitivity to fungicides, this time, the authors indicated that this would not be relevant to the present work. I will insist on this request since Cu and Fe, throughout the manuscript, are referred to as metal inhibitors of fungi. I still think it is very relevant because it would bring substantial novelty to the work: as mentioned in the discussion, both copper and iron in high concentrations are toxic to most organisms, and therefore, it was not surprising that relatively high concentrations of both metals, alone or in combination, would have a deadly effect on Botrytis cinerea. Again, this point is of significant importance and should add novelty to the reported results. To provide an additional example of the problem of the originality of the obtained results, please refer to the following lines: 300-302 and 309-311. In both sections, results are compared to previously published data, in the results section, not in the discussion section of the manuscript. As the authors wrote, this example indicates that similar results for Botrytis have been published elsewhere. The suggested additional (and simple) experiment is essential to improve this feature of the MS.

From a methodological point of view, at least one critical questions need to be answered. In most virulence assays of B. cinerea, wounded tissue is used then the Botrytis strains (or mutants) have a defect of plant tissue penetration, but this is not the case with the B05.10 strain used in this work (Figure 9). Therefore, the authors need to give a convincing explanation for the need to use wounded strawberries in particular since these fruits are highly susceptible to Botrytis attack. Another additional reason is requested for the virulence assay performed in tomato leaves.

In line 179, it is unclear from the specified information if the phylogenetic tree was constructed from individual genes or if an in-silico concatenated sequence was used. Only when you read the results section, you can infer that the former was the case. Therefore, it is crucial to re-write the methods section more appropriately.

Although the authors indicated that the language was edited, there is a significant space for essential improvements. For example, please go to lines 115-119. In this section, the authors talk about the effect of copper on two fungi (two species are mentioned), but only one IC50 is provided. Moreover, they indicate that Cu may bind on spores, but: on which fungi? In line 119, the following sentence is written: “The mechanism of action of Cu on this pathogen…”. Again, they are talking about two fungi, not one fungal specie. Please, clarify and improve the language.

The following errors were still identified in the manuscript after a second round of revisions:

Line 37: B. cinerea and not de full name
Line 49: there is an underscore symbol
Line 54: boscalid not Boscalid
Line 66-67: the same is true when present in very low concentrations. Please, clarify the sentence.
Line 82: Please, correct the first citation (it has more than two authors).
Line 89: replace “including fungi” by “including B. cinerea”
Line 101: Which redox properties?
Line 157: remove “agar” since it is already included in “MEA”
Line 164-165: remove “of our laboratory according”
Line 167-170: the “(“ and “)” symbols should not be in italics.
Line 169: NEP1 is a single gene. Therefore, “protein” not “proteins”.
Line 171: Which regions? Please, clarify. Do you mean that more than one region was amplified for each gene. Why did the authors not amplify the entire ORF?
Line 174: Regarding Staats et al., is it 2005 or 2007?
Line 175: “to amplify the boty transposon”
Line 176: “to detect the flipper transposon”
Line 177: at the end of the sentence authors should include “, respectively”.
Line 289: “was validated”
Line 338: remove “work from our laboratory”
Line 351: the name of the specie is incorrectly written
Lines 387-388: please, English editing is needed
Lines 458-459: incorrectly cited

Experimental design

Please, refer to the former section regarding experimental questions.

Validity of the findings

Please, go to the "Basic reporting" section regarding novelty of the results. A simple yet powerful experiment is requested.

---

## Round 0.3 · Minor Revisions

Both reviewers have issues that I think where polished in this version according to the rebuttal letter and the corrected text. Please check that the reading of the manuscript goes smoothly and the quality of all figures.

---

## Round 0.4 · accepted · Accept

We thank the authors for the careful consideration of all the suggestions and the improvement of their manuscript. It is ready for publication. Greetings.